# Development and characterization of an oral microbiome transplant among Australians for the treatment of dental caries and periodontal disease: A study protocol

Sonia Nath[1]*, Peter Zilm[2], Lisa Jamieson[1], Kostas Kapellas[1], Nirmal Goswami[3], Kevin Ketagoda[2], Laura S. Weyrich[4,5]*

1 Australian Research Centre for Population Oral Health, Adelaide Dental School, The University of Adelaide, SA, Australia, 2 Oral Microbiology Laboratory, Adelaide Dental School, The University of Adelaide, Adelaide, SA, Australia, 3 Materials Chemistry Department, CSIR-Institute of Minerals and Materials Technology, Acharya Vihar, Bhubaneswar, India, 4 Department of Anthropology and the Huck Institutes of the Life Sciences, The Pennsylvania State University, University Park, PA, United States of America, 5 Australian Centre for Ancient DNA, School of Biological Sciences and the Environment Institute, University of Adelaide, Adelaide, SA, Australia

* sonia.nath@adelaide.edu.au (SN); Lsw132@psu.edu.au (LSW)

**Funding:** The study has been funded: Author: LW Funder: National Health and Medical Research Council (NHMRC) Project grant number:

## Abstract

### Background

Oral microbiome transplantation (OMT) is a novel concept of introducing health-associated oral microbiota into the oral cavity of a diseased patient. The premise is to reverse the state of oral dysbiosis, and restore the ecological balance to maintain a stable homeostasis with the host immune system. This study will assess the effectiveness, feasibility, and safety of OMT using an interdisciplinary approach.

### Methods/Design

To find donors suitable for microbial transplantation, supragingival plaque samples will be collected from 600 healthy participants. Each sample (200μL) will subsequently be examined in two ways: 1) 100μL of the sample will undergo high-throughput 16S rRNA gene amplicon sequencing and shotgun sequencing to identify the composition and characterisation of a healthy supragingival microbiome, 2) the remaining 100μL of the plaque sample will be mixed with 25% artificial saliva medium and inoculated into a specialised *in-vitro* flow cell model containing a hydroxyapatite disk. To obtain sufficient donor plaque, the samples would be grown for 14 days and further analysed microscopically and sequenced to examine and confirm the growth and survival of the microbiota. Samples with the healthiest microbiota would then be incorporated in a hydrogel delivery vehicle to enable transplantation of the donor oral microbiota. The third step would be to test the effectiveness of OMT in caries and periodontitis animal models for efficacy and safety for the treatment of oral diseases.

APP1187737. URL: https://www.nhmrc.gov.au/funding The funders had and will not have a role in study design, data collection, and analysis, decision to publish, or preparation of the manuscript.

## Discussion

If OMTs are found to be successful, it can form a new treatment method for common oral diseases such as dental caries and periodontitis. OMTs may have the potential to modulate the oral microbiota and shift the ecological imbalances to a healthier state.

## Background

### Microbiota are essential for human physiology and health

The human body has co-evolved with approximately a trillion microbes that inhabit different niches and create an adaptive ecosystem that can influence host physiology [1]. The microorganisms in the body, collectively called microbiota, contribute to 200 grams of total body weight of an average person weighing 70 kg [2]. Microbiota inside and outside the body are estimated to outnumber human somatic and germ cells, suggesting that we reframe the human body as a 'superorganism'–an aggregation of the human body and the microbiome as one [3]. Microbiota contribute to host health by modulating the immune system, maintaining homeostasis, and defending against infection.

### Oral microbiome in dental caries and periodontal disease

The oral cavity harbours the second most complex microbial community in the body, followed by the colon [4]. The oral microbiome–the microorganisms and their environmental context–consists of 484 out of 775 bacterial taxa that are exclusively found in the oral cavity, according to the expanded Human Oral Microbiome Database (eHOMD) [5]. The tooth surface is the only non-shedding surface in the oral cavity compared to other areas in the mouth, and this facilitates a stable anchoring environment for growth and development of the microbiota. The supragingival and subgingival microbiomes are two of these key ecosystems and are comprised of diverse biofilms, but differ substantially in composition. The supragingival microbiome is demarcated by bacteria, such as *Streptococcus*, *Actinomyces*, *Haemophilus*, *Neisseria*, *Veillonella*, *Fusobacterium*, *Oribacterium*, and *Rothia* [6, 7], while the subgingival microbiome is predominated by *Treponema denticola*, *Tannerella forsythia*, *Aggregatibacter actinomycetemcomitans*, *Fretibacterium*, *Porphyromonas*, *Peptococcus*, *Filifactor*, and *Mycoplasma* [8].

Dysbiosis is a term utilized to describe a perturbation in the microbiome that departs from an otherwise balanced ecology to prolong, exacerbate, or induce detrimental health [1]. While the use of the term can by problematic [9] dysbiosis in the oral microbiome is described in numerous health conditions, notably dental caries and periodontal disease. Typically, dental caries is linked to dysbiosis of the supragingival microbiome, while subgingival microbiome dysbiosis may lead to periodontal disease [7, 10], although dysbiosis in other oral microbiotas (such as saliva) have also been linked to these oral diseases [11].

Dental caries is a dynamic, microbial biofilm-mediated, multifactorial, sucrose-driven disease that results from an unbalance in a phasic demineralization and remineralisation, which can subsequently cause demineralization of hard dental tissues [12]. Cariogenic bacteria colonizing the oral cavity are influenced by diet [13], as frequent intake of carbohydrate (mainly sucrose) can disrupt the ecology of this community by the selection of acidogenic and acid tolerant species that are responsible for caries development [12]. Previous research suggested that the primary pathogen for dental caries in children and adults was *Streptococcus mutans* [14]. While this pathogen certainly contributes to caries formation in some people, recent advances

in High Throughput Sequencing (HTS) of the 16S ribosomal RNA encoding gene have identified additional microbes linked to shifts in non-mutans streptococci, such as *Streptococcus sanguinis*, *Streptococcus oralis*, and *Streptococcus mitis*, as the initial colonizers [7, 15], as well as *Lactobacilli*, *Actinomyces*, *Bifidobacteria*, and yeast species [7].

Periodontal disease is a multifactorial, polymicrobial disease resulting in periodontal pocket formation, clinical attachment loss, bone loss, and subsequent leading to loss of tooth structure. In the transition from health to periodontal disease, there is an emergence of low abundant species and increased species diversity [10]. According to Socransky's model, the bacteria associated with periodontal disease belong to the so-called red complex, comprising *P. gingivalis*, *Tannerella forsythia*, and *Treponema denticola* [16]. However, as with dental caries, additional organisms have been identified using HTS. Bacteria associated with periodontitis are now considered to include distinct phyla (*Spirochetes* and *Bacteroidetes*), genera (*Treponema*, *Synergistes*, *Prevotella*, *Synergistes*, *Megaspaera*, *Selenomonas* and *Desulfobulbus*), and previously uncultivable species, including Gram-positive *Filifactor alocis* and *Peptostreptococcus stomatis* [17].

## Conventional treatment of oral diseases

Conventional methods for both the treatment dental caries and periodontal disease involves causative therapy. The treatment of dental caries involves removal of demineralised tissue and replacing with filling material [12]. The primary treatment for periodontal disease is mechanical removal of the plaque biofilm and management of the potential risk factors (for example, better glycaemic control in diabetic patients or reduction in smoking for active tobacco smokers). Although current methods are effective in managing dental caries and periodontal disease, they fall short of completely preventing and arresting the diseases before they occur, resulting in potential short term benefits and the likely re-emergence of disease [18]. The failure rates of dental restorative materials are high, at an annual rate of 8% due to secondary caries and bulk fracture [19]. Earlier interventional options, such as application of resin-based sealants on permanent molars, reduce caries by only 11%-51% [20] for limited time period of up to 48 months [21]. In the case of periodontal disease, almost all on periodontal maintenance therapy continue to show signs of progressive clinical attachment loss, with one to two thirds of people losing at least one tooth during an extended period of periodontal maintenance care [22]. It is clearly imperative to find necessary ways for better preventing and managing periodontitis and dental caries. This is especially true considering higher rates of tooth loss in ageing populations [23] and severe caries rates of disease in young children [24]. Oral microbiome therapy (OMT) may offer an effective alternative for the treatment of common oral diseases.

## Developing oral microbiome transplantation therapy in Australia

Oral microbiome transplantation (OMT) is a similar concept to Faecal Microbiome Transplant, which involves transferring oral bacteria from a healthy donor to a patient suffering from a disease, such as dental caries or periodontitis. OMT therapy has recently been successfully performed among dogs with naturally occurring periodontitis [25]; this successful procedure in canines involved placing healthy donor plaque samples into a sterile saline solution, loading the mixture on a nylon swab, and swabbing/irrigating the tooth surface of the transplant recipient. While this procedure has been proposed in humans using an array of techniques [26, 27], this procedure has not yet been completed in humans, and the most appropriate, effective approach has not yet been identified. Direct transplantation in humans from a donor to recipient requires logistic considerations and may require the identity of the

donor and recipient to be revealed to one another. It is also unclear if the irrigation method could deliver enough microorganisms for a successful OMT transfer.

To overcome this, we propose a novel approach to developing and testing OMT prior to human clinical trials. Overall, our approach involves (1) identifying and scrutinizing healthy donor plaque material; (2) growing the plaque microbiota *in vitro* in a flow cell, developed by extending the work of several previous teams [25, 26, 28, 29]; (3) integrating donor oral microbiota into a formulated hydrogel; and (4) transplanting donor material into an animal model of oral disease to test its efficacy and safety.

A key aspect of this study is the identification of healthy donors. The microbiome of a 'healthy' person shows a large degree of interpersonal diversity and finding a single 'healthy' state across all individuals is unlikely [10], making donor identification a complex process. However, concept of a 'super donor' is derived from faecal microbiome transplantation [30] and describes faecal donor material that results in significantly more successful outcomes than the stools of other donors; for example, the super-donor effect has been observed in various clinical trials for the treatment of inflammatory bowel disease [31, 32]. In the gut, 'super donor' microbiota is largely associated with the presence of keystone species or high diversity–features that are not necessarily linked to health improvements in the mouth (e.g., keystone species, such as *P.gingivalis* can be considered an oral pathogen [33, 34] and a higher diversity in oral microbiota can be linked with disease, such as periodontal disease [34]). In canine OMT, the authors confirmed that the oral microbiome of healthy dogs/participant was different from the oral microbiome of both periodontitis [25, 27] and edentulous patients [27] but failed to identify the constituents that demarcate a 'healthy donor'. Therefore, we propose that microbiota from an OMT super donor should have the following characteristics: (1) possesses limited abundances of 'red' complex microbes, as defined by Socransky, et al. [16] and known cariogenic species (*e.g.*, *Streptococcus mutans*) [35]; (2) has no transmissible or infectious pathogens (herpes virus, human immunodeficiency virus, etc.); (3) successfully grow in *in vitro* model, exhibiting higher proportion of live bacterial cells than dead bacterial cells; (4) be able to effectively grow in a host by causing a shift in the microbiome profile of an individual towards that of a donor; and (5) is efficacious against oral disease in a rodent model (*e.g.*, reduces inflammation) with limited negative outcomes (*e.g.*, oral infection or increases in oral disease markers). While effective 'super donors' need to be further examined during human OMT trials, we hope that these characteristics of donor microbiota will likely improve the chances of OMT success in humans. Nevertheless, additional safety, ethical, and pragmatic considerations need to be examined before OMT therapy is utilized in humans, and this study will provide foundational data necessary to determine if and how human OMT clinical trials should proceed.

## Study aims

The specific aims of the study are as follows:

Aim 1: Describing and identifying the healthy donor and testing the oral biofilm growth using a novel *in vitro* flow cell model.

Aim 2: Develop a delivery system for OMT therapy, including hydrogel applications:

Aim 3: Test oral microbiota transplant in rodent models:

　a. Examine the safety and efficacy of OMT of human microbiota in healthy rodents.

　b. Examine the safety and efficacy of OMT in a rodent caries and periodontal disease model.

## Methods

This study will be conducted in three stages according to each aim (Fig 1).

### Stage 1: Describing and identifying the healthy donor and testing the oral biofilm growth using a novel *in vitro* flow cell model

Supragingival plaque samples will be collected from all healthy participants, as identified during the oral screening process. A part of the samples (100 µl) will be stored at -80 C for downstream cultivation and further sequenced using amplicon and shotgun approaches. The other part (100 µl) of the plaque sample immediately post-collection will be grown *in vitro* on hydroxyapatite discs using a flow cell model for 14 days. The discs would be extracted after 14 days and further assessed microscopically and sequenced through HTS techniques.

### Step 1: Describing the oral plaque microbiome of healthy donors

**Ethics approval.**   The study has been approved by the University of Adelaide Human Research Ethics Committee (H-2020-34609) and complied with the Declaration of Helsinki for Medical Research involving Human Subjects. Participation information sheet will be given to each eligible participant (S1 File) and written informed consent will be obtained from participants before their involvement in the study (S2 File).

**Sample size calculation.**   It is often challenging to calculate power for microbiome analysis because the true effect size is usually unknown and the relationship between the microbiota community structure and each factor (alpha or beta diversity) is highly dimensional. For our study sample size calculation was based on the methods suggested by Casals-Pascual et al.

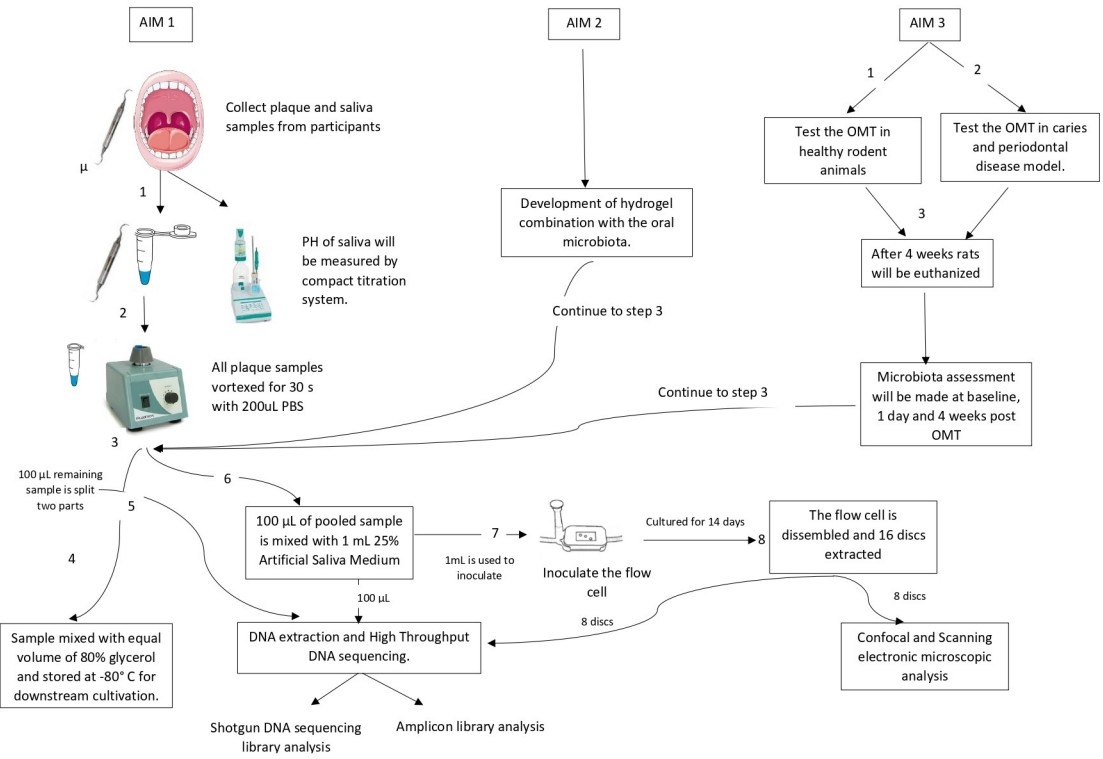

**Fig 1. Study flow diagram.**

(2020) [36]. The sample size was calculated using the mean and standard deviation for alpha diversity (species diversity in a community) derived by Burcham et al. (2020) [37]. The minimum expected standard deviation was 2.53. Therefore, at an error of 0.05 and power of 80% the calculated sample size was 583. The sample size was rounded to the nearest 100 to account for loss of samples due to poor sequencing depth.

**Study population and recruitment.** 600 adults aged 18+ years will be recruited. Recruitment will occur in the greater Adelaide region (South Australia) via advertisements in social media and similar platforms. A passive snowballing technique will be used, recruited study participants would be asked to discuss the research project with their family, friends, peers and colleagues and this process will continue until the desired sample size is achieved. The patient interview, clinical and oral assessment, and dental plaque sampling will take place in dental clinics, Adelaide Dental School, University of Adelaide. The recruitment process will begin on June 2021.

**Inclusion and exclusion criteria.** Systemically and orally healthy adult's $\geq$ 18 years of age would be selected. The inclusion criteria are: (1) presence of all set of teeth with the exception of impacted/missing wisdom teeth or orthodontic extraction; (2) periodontally healthy participants having probing depth (PD) of $\leq$ 4 mm and no loss of clinical attachment; (3) presence of less than 20% of bleeding on probing (BOP); (4) score of 2 or below in the Caries Assessment Spectrum and Treatment (CAST) index [38].

The exclusion criteria includes: (1) participants with any systemic disease such as diabetes, cardiovascular condition (angina or heart failure), cancer, respiratory disorder, bone or joint disorder, epilepsy, gastro-intestinal disorder (ulcerative colitis or Crohn's disease) or kidney dysfunction; (2) pregnant or lactating mothers; (3) taking medication orally that may impact the oral microbiome in the last 3 months such as (i) antibiotics, anti-fungal, anti-viral, or anti-parasitic; (ii) corticosteroids (Prednisolone, Flonase, dexamethasone, Flovent); (iii) cytokines or drugs that can stimulate the immune system; (iv) methotrexate or other agents that can supress the immune system; (v) commercial probiotics (e.g. probiotic tablets or Yakult); (4) those who have undergone dental treatment (scaling or dental filling) in the last 3 months. Any persons who have ever tested positive for SARS-CoV-2 (COVID-19) or are having any flu like symptoms like fever, cough, shortness of breath or a recent travel history interstate or internationally would additionally be excluded.

**Data collection.** In view of the COVID-19 pandemic, strict sterilization and personal protection measures will be taken while clinically examining the participant and plaque/saliva sampling to reduce exposure to body. All research staff would wear personal protective equipment including masks and non-latex sterile gloves while conducting oral examinations.

i. Questionnaire: Data will be collected using a two self-report questionnaire. The questionnaire will be offered in both paper and online formats, depending upon participant preference. A pretested custom questionnaire has been formed by the authors to capture the socio-demographic data using the REDCap® (Research Electronic Data Capture) software [39]. The questionnaire will collect information on demographics, details on smoking and alcohol consumption, general health status, physical activity, oral health behaviour and emotional well-being (S3 File). The second questionnaire, the Dietary Questionnaire for Epidemiological Studies Version 3.2 (DQES v3.2) by the Cancer Council Victoria will be used to measure diet and nutritional habits [40].

ii. Anthropometric measurements: The height and weight of the participants will be taken and body mass index would be calculated.

iii. Clinical oral examination: The oral assessment will be done with study participant laying in a supine position whilst a registered dental clinician examines the hard and soft tissues of their mouth. BOP will be recorded as the overall percentage of sites with bleeding occurring within 15 seconds of probing. PD will be measured from the free gingival margin to the base of the pocket at the selected sites using a UNC-15 periodontal probe (Hu Friedy, Chicago, IL, USA).

For caries assessment, the CAST index will be used. After removing excess saliva with cotton rolls or gauze, the entire dentition would be examined with a mouth mirror and probe and a score from 0–9 will be given. Participants with code 0 (No visible evidence of a distinct carious lesion), 1 (Pits and/or fissure are at least partially covered with a sealant material), and 2 (A cavity is restored with an (in) direct restorative material) will be considered as "healthy participants", and a higher score would not be accepted.

iv. Dental plaque collection: On the day sample collection, the participants would be advised to abstain from brushing their teeth and using any form of mouthwash. Supragingival plaque samples would be collected from four sites (buccal surface of maxillary central incisor and lingual surface of mandibular central incisor and mesio-buccal surface of maxillary molar and mesio-buccal surface of mandibular molar) using sterile dental curette. The pooled plaque sample would be vortexed in 200μL of phosphate buffer solution and then samples would then be immediately transported to the Oral Microbiology Laboratory for inoculation in to the flow cells.

v. Stimulated whole saliva collection: The collection would be conducted according to a method previously described [41]. The participant would be instructed to sit upright motionless and instructed to chew an inert gum. The participant would be asked to spit everything into the graduated collection tube. The fist two minute collection would be discarded, and then the next five minute collection would be retained for further analysis. A salivary flow rate would be caluclated based on the flow rate per min.

Additonally, the pH of the saliva will be determined using a saliva titration system. The level of acid produced will be assessed by flowing the saliva sample by adding 8ml of 10% glucose and incubating for 2 hours. A pH of 6.8 will be maintained by the titration of 0.1M NaOH.

**Splitting of samples.** The pooled plaque samples would be vortexed for 30s at low speed in 200 μl of phosphate buffer solution (PBS) for homogenization. A part of the sample (100μl) will be stored at -80°C and processed later for DNA extraction and high throughput sequencing and the second part (100μl) would be used for inoculation in the flow cell.

## Step 2: Inoculation and growth of the *in vitro* biofilm

The sample (100 μl) would be mixed with 1 ml of 25% modified version of Artificial Salivary Medium (ASM) before inoculation in the flow cell (Fig 2). The ASM includes the following components: 0.50 g/L tryptone (Oxoid, Hampshire, England), 0.50 g/L neutralised bacteriological peptone (Oxoid), 0.625 g/L type III porcine gastric mucin (Sigma-Aldrich, Steinheim, Germany), 0.25 g/L yeast extract (Oxoid), 0.05 g/L KCl, 0.05 g/L $CaCl_2$, 0.088 g/L NaCl, and 1mg/L haemin (Sigma-Aldrich). The solution was supplemented with 2.5mM DTT (Sigma-Aldrich) and the pH was adjusted to 7.0 [42]. The sample would be injected into the inoculation port (Fig 2F) then be cultured on hydroxyapatite (HA) discs (Clarkson Chromatography Products Inc. PA, USA) in 3D-printed flow cells for a period of 14 days. The flow cell would remain static for 12h to ensure each HA discs would be coated with ASM, as this would facilitate the formation of acquired pellicle for bacterial adhesion. Based on our unpublished preliminary

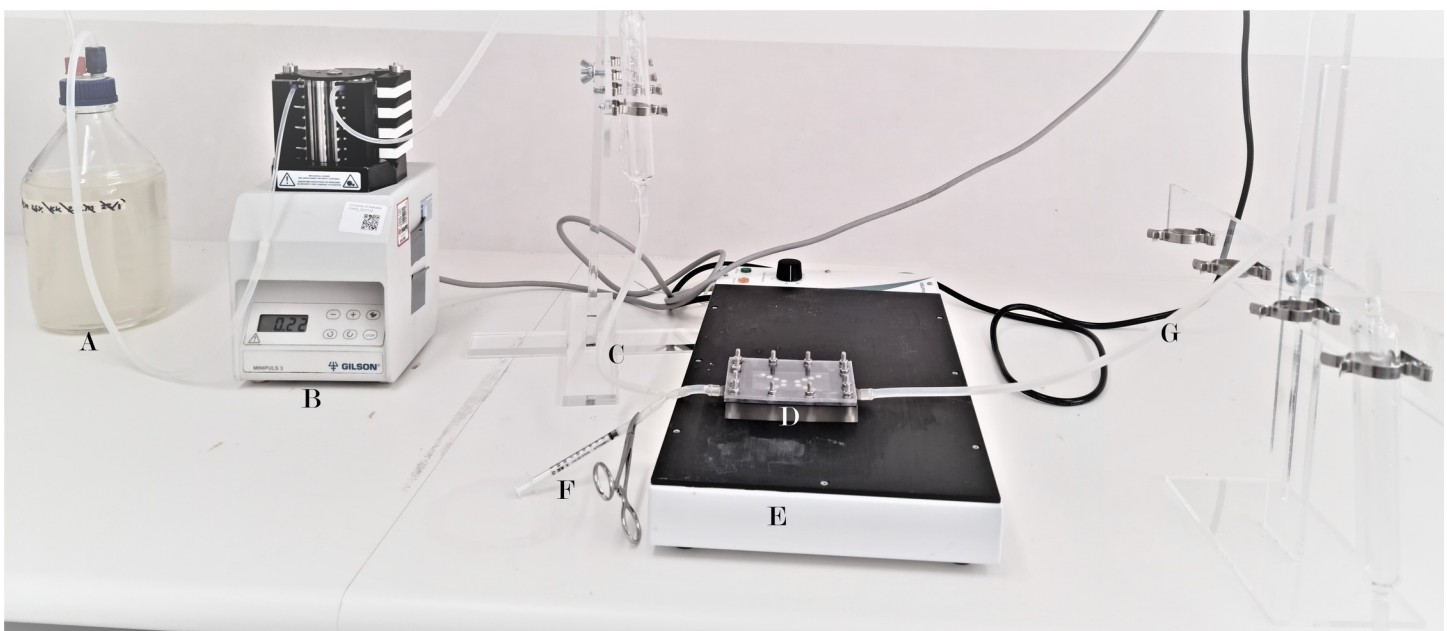

**Fig 2.** The flow cell and its components: The components of the system included; A medium reservoir (A), a multi-channel peristaltic pump (B), a break line (C), the flow cell (D), a slide heating tray (E), inoculation port (F), expelled into waste container (G).

data, each flow cell will produce 16 discs with ~$1.15 \times 10^4$ live cells on each. After 14 days, the flow cell would be dissembled and all the 16 discs would be extracted. Half of the discs would be used for microscopic examination and the other half would be stored at -80˚ C until used for amplicon and shotgun sequencing.

**Microscopic examination.** Confocal Laser Scanning Microscopy (CLSM) and Scanning Electronic Microscopy (SEM) will be used to visualise the microbial communities grown on the eight HA discs. Microscopic examination is essential to ensure that biofilms are actively growing. CLSM semi-quantitatively analyses the proportion of living:dead microbes using BacLight® LIVE/DEAD staining and imaging using IMARIS® software [43]. SEM will determine the morphological structure of the biofilm.

**DNA extraction and sequencing.** DNA from all samples (PBS plaque sample and eight discs) and negative controls will be extracted using a DNeasy PowerSoil Pro Kit®. Amplicon and Shotgun will be employed to: (1) study the overall bacterial population structure and compare communities to existing data sets on healthy individuals (amplicon) and (2) identify specific species and characterize their functions (shotgun).

Amplicon Library analysis: Amplicon sequencing will be done by amplifying the 16S rRNA encoding the $V_4$ gene region with region-specific primers, as previously described [44]. Amplicon libraries will then be cleaned, quantified, pooled, and custom sequenced in parallel to a level of >10, 000 sequences per sample on an Illumina MiSeq [44, 45]. The QIIME2 software will be used for analysis of all amplicon data [46]. In QIIME2, amplicon sequence variants (ASVs) will be taxonomically identified by comparisons to eHOMD and the SILVA databases. Alpha diversity (observed species and Shannon's Diversity Index) and beta-diversity (unweighted UniFrac and Bray-Curtis indices) will be analysed. The donor microbiome will also be compared to healthy supragingival plaque specimens publicly available on QIITA, which is a web-based microbiome comparison platform that allows users to rapidly compare their data with Human Microbiome Project (HMP) and other submitted data sets [47].

Shotgun DNA sequencing library analysis: Shotgun sequencing is random sequencing across entire genomes [48]. To prepare shotgun libraries, custom DNA adaptors with sample specific barcodes will be ligated to random DNA fragments within the samples, as previously described [45]. After cleaning and pooling, shotgun libraries will be sequenced to at least 100 million sequences per sample on an Illumina NovaSeq. To identify bacterial species and functions in the shotgun data, we will map and remove all human sequences and then compare the remaining sequences to a custom curated RefSeq and HOMD genome database and KEGG protein predictions in two analytical pipelines: (1) the MetaPhylan and HUMAnN (2) and the MALT and MEGAN6 CE analysis pipeline packages [45]. This will enable the team to examine links between species and their functions. Bacterial genome assembly using GroopM will be done to link functions to certain species [49].

## Aim 2: Develop a delivery system for OMT therapy

To develop hydrogels as an OMT delivery vehicle, hydrogels comprising of dopamine and oxidized sodium alginate will be prepared in a multi-step process. First, sodium alginate will be oxidized by sodium periodate in an ethanol-water (1:1 v/v) mixture. Dopamine (DA) will then be grafted to the oxidized sodium alginate following a well-established bioconjugation method. Specifically, the amine group of the dopamine will be conjugated to the carboxylic acid group of oxidized sodium alginate using EDC (1-(3-(Dimethylamino) propyl)-3-ethylcarbodiimide hydrochloride) and NHS (N-hydroxysuccinimide) as a coupling agent. The product will be purified by extensive dialysis and used for the hydrogel preparation.

For the hydrogel preparation, dopamine grafted oxidized sodium alginate will be first mixed with acrylamide (AM) followed by the addition of ammonium persulfate (APS), $N,N'$-methylenebisacrylamide (BIS), and $N,N,N',N'$ tetramethylethylenediamine in an inert ($N_2$) atmosphere. Inert atmosphere is necessary to prevent the over oxidation of dopamine. A transparent hydrogel will be formed after 4 hours. The composition of the hydrogel can be varied by changing the amount of dopamine grafted oxidized sodium alginate in the final solution.

In addition, hydrogel containing polydopamine (PDA) and polyacrylamide (PAM) will also be prepared as an alternative. In such case, dopamine will be polymerized first in the presence of sodium hydroxide followed by the sequential addition of AM, APS, and BIS under mechanical stirring in an ice bath. PDA-PAM single network hydrogel will be formed after 10 min. DA/AM ratio will be varied to optimize the properties of the hydrogels.

Hydrogel mechanical stability will be assessed by determining their storage modulus (G') and the loss tangent (ratio of loss modulus G" to storage modulus G'). The chemical characterization of the hydrogel will be assessed by FT-IR spectroscopy, XPS, and thermo gravimetric analysis. The morphological characterization of the hydrogel will be performed by SEM and atomic force microscopy (AFM). After successful characterization, microbes will be added into the hydrogel, and their survival monitored using microscopy and sequenced (as described above) by every 12 hours for a three-day period, reminiscent of the transport and application time needed for OMT therapy [50]. However, the efficacy of this application method still needs to be examined in murine models, and if we experience unforeseen issues with the hydrogel approach, other strategies will be explored, such as application of plaque microbiota in buffers (e.g., as the canine OMT study was done), varnishes, or mouthwashes.

## Aim 3: To test the oral microbiome transplant in rodent models

**Ethics.** Before the start of the study, ethical approval will be obtained from the University of Adelaide Animal Ethics Committee and comply with National Health and Research Council (Australia) Code of Practice for Animal Care in Research and Training (M-2021-022).

**Animals.** 6–8 weeks old, Sprague-Dawley rats and Balb/c mice will be used in healthy trials of OMT. The animals will be obtained from the Laboratory Animal Services of the University of Adelaide and will be housed in animal holding facilities. All animals will be subjected to 5 days of acclimatization and be housed together to promote amelioration of the microbiota between animals. Animals would be caged in screen-bottomed cages, as this prevents accumulation of faeces and urine in the cage. Bedding for the animals will not be used as the animals might ingest it, which can lead to impaction in fissures and interproximal areas [51]. The animals' diet will comprise powdered, sterile, non-granular food to prevent impaction of food around the gingiva and they would have access to sterile non-acidic water *ad libitum*. They will be kept in a room with a 12-hr light/dark cycle with temperature ranging from 22 to 24°C [52].

## Aim 3a) Examine the safety and efficacy of OMT of human microbiome in healthy mouse/rats

**Experimental design.** Two groups per experiment will be repeated with six different donor microbiotas:

1. *Control group 1*: Placement of hydrogel (without bacteria) once daily for three days on tongue, buccal areas, gingiva, palate, and tooth surfaces.

2. *Experimental group 1*: OMT (donor microbiota in hydrogel) once daily for three days, as above.

Microbiota assessments will be made at baseline, 1 day after the final OMT treatment, and 4-weeks post-OMT, using microscopy and DNA sequencing, as described above. A behavioural and physical health check of the animals will be done weekly and at the end of the study. The animals would be replaced if there were any deviation from normal healthy behaviour. To collect microbiota during the experiment, the oral cavity of an anesthetized mouth will be swabbed for 30 sec using nylon swabs, over the tongue, buccal areas, gingiva, palate, and tooth surfaces. Swabs will be placed into sterile PBS buffer, alongside a negative control, and analysed immediately or frozen at -80C until DNA extraction. At the end, rodents will be swabbed a final time, and concluding oral samples taken for microscopy and DNA analysis.

## Aim 3b) Examine safety and efficacy of OMT in caries and periodontal disease model

**Caries model.** Sprague-Dawley rats will be used and screened for the presence of *Streptococcus rattus* (analogous to *Streptococcus mutans)* by polymerase chain reaction. The animals will be fed the NIH diet 2000 [53] containing either 0% sucrose (non-cariogenic) or 56% sucrose (cariogenic), which are fed by program feeding (5 times/day) to reduce artefacts due to eating frequency. The rodents will be intermixed weekly and co-housed for 4 weeks to enable adjustment to the environment and ameliorate their microbiota.

**Experimental design.** Eight 10-day-old rats from different litters will be evenly distributed amongst four groups and monitored for 4 weeks.

1. *Control group 1*: no bacteria but fed non-cariogenic diet.

2. *Control group 2*: no bacteria but fed cariogenic diet.

3. *Experimental group 1*: OMT once daily for three days with non-cariogenic diet.

4. *Experimental group 2*: OMT once daily for three days with cariogenic diet.

Before the transplantation of the oral microbiome, full mouth debridement would be done for all the rats in experimental group followed by oral irrigation with 0.1% of sodium hypochlorite. The sodium hypochlorite would be buffered with a neutralising agent [27]. Caries will be scored using a method described by Keyes [54] and Larson [55], which provides information on the lesion sites and severity. After 4 weeks, the rats will be humanely euthanized by carbon dioxide chamber ($CO_2$) for 5–7 min. The maxilla and mandible will be harvested by sharp dissection and de-fleshed mechanically. The jaws would be fixed in 10% formalin and 1% sodium hypochlorite solution for a minimum 48 hours before defleshing. Scored jaws will be kept to ensure consistency in scoring between three biological replicates. Microbiota will be assessed at baseline, 1 day post- OMT, and at the end of 4 weeks, as described in Aim 3a.

**Periodontitis model.** Experimental periodontitis will be induced in 5–7 week old BALB/c mice after an initial 5 days being fed with either control or experimental diet and receiving kanamycin (1mg/mL) ad libitum for a 7-day period in their drinking water to supress the indigenous oral microbiota [56]. For induction of experimental periodontitis, the mice will be inoculated with either the suspension vehicle only (2% carboxyl-methylcellulose) or *Fusobacterium nucleatum* and *Porphyromonas gingivalis* in the periodontium for 5 weeks twice weekly. The inoculation will consist of 100 μL of bacteria either $10^{10}$ colony-forming units/mL of *P. gingivalis* alone or $5 \times 10^9$ colony-forming units/mL of *P.gingivalis* and $5 \times 10^9$ colony-forming units/mL of *F.nucleatum* suspended in 2% carboxymethyl cellulose [57]. Experimental mice will receive the OMT one week after *F.nucleatum* and *P.gingivalis* inoculation. Each group will consist of eight mice from different litters:

1. *Control group 1*: orally inoculated with the suspension vehicle.

2. *Control group 2*: mice inoculated with *F.nucleatum* and *P.gingivalis*.

3. *Experimental group 1*: Inoculated with suspension vehicle with OMT once daily for 3 days.

4. *Experimental group 2*: *F.nucleatum* and *P.gingivalis* inoculation followed by OMT once daily for 3 days.

To reduce the effect of periodontal inflammation all the mice would undergo full mouth scaling and polishing before oral microbiome transplantation on experimental group. To further supress the resident microbiota of the recipient mice, subgingival and oral irrigation with 0.1% sodium hypochlorite would be done for 5 min. The sodium hypochlorite would then be inactivated by using 23μM buffered sodium ascorbate for 10 min [25, 27]. The OMT would be delivered locally in hydrogel loaded dental tray. The loss of alveolar bone will be measured prior to OMTs/infections and throughout the experiment by live animal computed tomography scanner (Skyscan 1076 High Resolution *In Vivo* Scanner; Skyscan, Kontich, Belgium) measuring changes in bone volume from the cemento-enamel junction to the alveolar bone crest in the jaws [58]. The scanner has a 10 megapixel camera with a tungsten 100 kV X-ray source and a spot size of 5 um. For scanning, the X-ray source will be operated at 75 kV X-ray source and 120 mA with a 1-mm tungsten filter. The scanning width would be set to 35 mm with a pixel of 9 um pixels [58].

After 5 weeks, the rats will be humanely euthanized by $CO_2$ and the jaws would be dissected to similar procedure described as above for caries model. Alveolar bone loss will be assessed using a Leica MZI6FA stereomicroscope (Leica Microsystems, Wetzlar, Germany) compared to controls [57]. Microbiome will be assessed by microscopy and high-throughput DNA sequencing.

All the data will be compared using ANOVA, and experiments will consist of three biological replicates. For safety of OMT, the presence and absence of discomfort would be measured

by monitoring the vital parameters and any adverse reactions of the oral mucosa like redness, inflammation, bleeding, friability of the tissues to the transplants would be recorded separately.

## Discussion

To the best of our knowledge, OMT for the treatment and prevention of oral diseases has not been established anywhere in the world. OMT therapy may provide alternative tool to overcome poly-microbial oral diseases that are typically recalcitrant to treatment or prevention such as dental caries and periodontal disease. Our approach represents several strategic advantages at each step.

First, the identification and use of "super donors" in OMT therapy is likely to be a key component for successful OMT. To identify potential "super donors" for OMT, donors must undergo a rigorous selection process. As proposed in our protocol, our proposed screening process involves collecting a detailed medical history and analysis of the microbiome. Any participant under any medication for systemic illness, undergoing treatment, or taking drugs that might have an effect on the immune or inflammatory mechanism would be excluded. This could lead to rejection of high number of participants as observed by many studies [59]. The microbiota from a potential 'super donor' will be placed in a biome bank and would be brought to clinical use when needed. Using 'super donors' in OMT development should reduce costs, as large number of samples can be stored and processes standardised for safety, traceability, and monitoring. A search for an OMT 'super donor' also have further benefits for the oral microbiome research community. Our study will provide the baseline data of composition and characteristics of healthy oral microbiome among Australian adults. Through our questionnaire we are recording many intrinsic host-related factor that are known to affect the oral microbiome, such as diet, physical exercise, oral hygiene habits, access to dental care, socio-economic status and history of smoking and alcohol consumption [60]. We are measuring the buffering capacity of the saliva of each donor and this would help us understand if there is any correlation between species of the oral microflora and favourable pH (6.75–7.25) of the oral cavity [60]. High-throughput DNA sequencing approaches will describe the composition, species, and functions of an oral healthy microbiome among Australians and correlate with genetic background of the donor. Several researchers have examined the healthy oral microbiome only in the context of disease, with little exploration into the function and species that are demonstrative of protective and beneficial oral microbiome [4, 6, 7, 61].

Second, the use of *in vitro* flow cell model is advantageous for several reasons: 1) large amounts of microbiota from a single donor could be mass-produced; 2) biofilms can be screened for pathogens or potentially unwanted species prior to administration; and 3) microbial communities can be manipulated *in vitro* to improve efficacy (*e.g.*, adding probiotics or removing certain problematic species). Further, an effective *in vitro* growth system of dental plaque also provides opportunities to test biofilm removal and competition, which is critical to examine if or how a recipient's microbiota may need to be pre-treated before receiving an OMT (i.e., pre-treatment via physical or chemical removal, such as antibiotics). Lastly, this *in vitro* growth strategy could also potentially be adapted to examine unique oral environments. For example, we used 25% mucin rich ASM to mimic a healthy mouth and cultivate a wide range of bacterial species, but serum, sucrose, or other metabolites could be added to mimic unique human diets and lifestyles and examine growth in specific oral environments. For future research, the *in vitro* model would allow us to genetically engineer the biofilm to selectively grow healthy species for transplantation.

Third, adhesive hydrogels are used to administer donor microbiota during OMTs. Adhesive hydrogels have been utilized in humans (e.g. local drug delivery for periodontal therapy and caries prevention [62, 63]), are non-toxic, and can be applied in the mouth for extended periods without irritation (e.g., overnight to 8 days) [64] representing a plausible delivery vehicle for donor microbiota during OMT. Previous research has shown that gram-positive and gram-negative bacterial cells can successfully adhere to hydrogel matrix by cross-linking to the hydrogel polymer chain [65–67]. Hydrogels could prove beneficial for OMT delivery because the hydrogel matrix mimics the extracellular polymer produced in a biofilm [65]. A variety of hydrogels has been used as a carrier material for local drug delivery for bioactive substances for the treatment of periodontitis and dental caries [62, 63, 68, 69]. Hydrogels have multifunctional properties such as structural integrity, slow and controlled release effect, pH- responsiveness, broad antimicrobial spectrum, tissue repair and adhesive properties [68]. Hydrogels can be easily processed into various shapes, especially the injectable and thermos-sensitive hydrogels are capable of filling up narrow spaces in the oral cavity such as gingival crevices and this would be suitable for the delivery of the oral microbiota [64, 69]. For example, hydrogels inoculated with donor microbiota could be easily applied either in a clinical setting or at home by placing the hydrogel in a dental tray, similar to at-home tooth whitening kits that are commercially available today, providing easy ways for application. The hydrogels loaded with oral microbiota could be seeded into a dental tray to deliver and maintain the hydrogel in the subgingival and peri-gingival area, providing a sustained opportunity to seed the microorganisms onto the surface of teeth to endure and overcome the existing selective pressures from the environment of the recipient's oral cavity [70, 71]. This approach provides a novel means to allow microbiota to be applied topically for a sustained period, which may be critical to achieve a sustained therapeutic effect.

Lastly, this approach utilized murine models to examine OMT safety and efficacy before proceeding with human trials; we selected murine models of oral disease that utilized microbes for disease initiation to ideally represent more natural disease states with dysbiosed microbiota. These experimental models for periodontitis and dental caries in rats and mouse have been very beneficial to our understanding of oral disease progression, providing reproducible radiographic, clinical, molecular and histologic features of human oral disease [51, 57]. Specifically, the bacterially induced experimental periodontitis model (oral gavage model) is important for the study the disease progression and host immune response and would be more suitable for the investigation of the pathophysiological factors induced by oral microorganism's dysbiosis. Repeated dual inoculation with *P.gingivalis* and *F.nucleatum* has previously shown to induce alveolar bone loss in an experimental periodontitis model [52, 57]. Alternative models of disease (i.e., silk-ligature models of periodontal disease [72]) could also be examined in the context of OMT efficacy and safety, perhaps providing unique insights into how shifts in immune response during OMT may be linked to health. However, examining if donor OMT can overcome known microorganisms linked to disease may improve our understanding of OMT efficacy in humans.

OMT therapy can be beneficial for the treatment of oral disease and has potential to cause a lifelong shift towards a healthy oral microbiome. Vertical transmission of bacteria has been found between mother and child and occurrence of similar periodontal and cariogenic pathogen [7, 73]. By introducing the therapy in the oral cavity at a younger age, they can benefit by overcoming the "unhealthy" microbiome that might be inherited from their parents. The application of OMT therapy during eruption of the permanent dentition may provide a positive change to person's oral microbiota for years afterward. In addition, the aging population has increasing levels of dental caries, and OMT applications with the introduction of dental prosthetics could alter the microbiome to a more stable and balanced microbiota. In Australia,

the prevalence of moderate and severe periodontitis is also increasing [23, 74]. Current treatment regimens are not successful in completely arresting the disease [75]. OMT therapy could be a potential solution to this problem. OMT could be applied to people at the early stages of disease to stop the progression of disease by creating an environment for the healthy microbiome to form. Australian remote and rural communities are significantly more affected by dental caries and periodontal disease [23], so OMT therapy may be beneficial for remote and rural communities that have limited access to dentists. This therapy could be administered at a key point in life to result in sustained oral microbiota alteration, limiting the need for expensive and time-consuming annual treatments.

## Conclusion

Our research provides a roadmap to produce preliminary results investigating the feasibility and efficacy of treating dental caries and periodontitis models using OMT. In addition, it will provide the opportunity to expand and understand the Australian oral microbiota diversity and function in human health and basic biofilm physiology. Nevertheless, additional safety and efficacy examinations will be needed before this therapy can be tested in human clinical trials.

## Ethics approval and consent to participate

The study has been externally peer reviewed and approved by the funding organization National Health and Medical Research Council (NHMRC), Project grant number APP1187737. The NHMRC is a statutory authority and the primary agency of the Australian Government responsible for medical and public health research. This study has been approved by the University of Adelaide Human Research Ethics Committee (H-2020-198). The participation is voluntary, patient information sheet detailing the study would be given to eligible study participants, and written informed consent will be obtained before their involvement in the study.

## Supporting information

**S1 File. Participant information sheet.** This would be distributed to every eligible study participant informing about the study. Consent would be obtained after the information sheet has been completely read and understood by the participant.
(DOCX)

**S2 File. Consent form.** The consent form would require patient's signature on agreeing to participate in the study.
(DOCX)

**S3 File. Questionnaire.** This is online questionnaire designed in REDCap.
(PDF)

## Author Contributions

**Conceptualization:** Peter Zilm, Laura S. Weyrich.

**Formal analysis:** Kevin Ketagoda.

**Funding acquisition:** Peter Zilm, Nirmal Goswami, Laura S. Weyrich.

**Investigation:** Nirmal Goswami.

**Methodology:** Sonia Nath, Peter Zilm, Kostas Kapellas, Nirmal Goswami, Laura S. Weyrich.

**Project administration:** Peter Zilm, Laura S. Weyrich.

**Supervision:** Peter Zilm, Lisa Jamieson, Kostas Kapellas, Laura S. Weyrich.

**Writing – original draft:** Sonia Nath, Laura S. Weyrich.

**Writing – review & editing:** Sonia Nath, Peter Zilm, Lisa Jamieson, Kostas Kapellas, Nirmal Goswami, Kevin Ketagoda, Laura S. Weyrich.

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
