## [Decision Letter · Decision Letter 0]

12 Jul 2021

PONE-D-21-16611

Development and Characterization of an Oral Microbiome Transplant Among Australians for the Treatment of Dental Caries and Periodontal Disease: A Study Protocol.

PLOS ONE

Dear Dr. Nath,

Thank you for submitting your manuscript to PLOS ONE. After careful consideration, we feel that it has merit but does not fully meet PLOS ONE’s publication criteria as it currently stands. Therefore, we invite you to submit a revised version of the manuscript that addresses the points raised during the review process.

In addition to clarification of the results throughout, both reviewers found the concept of the 'super donor' to be poorly described, if not ill conceived. Due to the plasiticity in microbiome between individuals, it is likely to the authors' benefit to more carefully consider this idea.

We look forward to receiving your revised manuscript.

Kind regards,

Christopher Staley, Ph.D.

Academic Editor

PLOS ONE

Journal Requirements:

2. Please update the Animal ethics statement to include the date when approval was received

Additional Editor Comments (if provided):

Reviewers' comments:

Reviewer's Responses to Questions

**Comments to the Author**

1. Does the manuscript provide a valid rationale for the proposed study, with clearly identified and justified research questions?

Reviewer #1: Partly

Reviewer #2: Yes

2. Is the protocol technically sound and planned in a manner that will lead to a meaningful outcome and allow testing the stated hypotheses?

Reviewer #1: Partly

Reviewer #2: Yes

3. Is the methodology feasible and described in sufficient detail to allow the work to be replicable?

Reviewer #1: Yes

Reviewer #2: Yes

4. Have the authors described where all data underlying the findings will be made available when the study is complete?

Reviewer #1: Yes

Reviewer #2: No

5. Is the manuscript presented in an intelligible fashion and written in standard English?

Reviewer #1: Yes

Reviewer #2: Yes

6. Review Comments to the Author

You may also provide optional suggestions and comments to authors that they might find helpful in planning their study.

Reviewer #1: This manuscript aims to develop and characterize an Oral Microbiome Transplant (OMT) for the treatment of caries and periodontal disease. Supragingival plaque samples would be collected from healthy participants to find donors suitable for microbial transplantation. Based on the current protocol, a novel in vitro flow cell model combined with microscopic examination and DNA sequencing were be used to describe the composition and functions of the donors. Hydrogels would be used to deliver microbiota for OMT. The effectiveness of OMT would be assessed using caries and periodontitis animal models. Although the current research would provide preliminary results of the feasibility of oral microbiome transplant, I still see some major areas of concern that I think need to be addressed before this manuscript is ready for primetime.

1. What is the specific definition of “super donor”? Authors mentioned that it is prudent to understand the variation and functions of a healthy oral microbiome and hence define a “super donor”, however, authors never presented data in support of any of these claims.

2. In Rodent periodontitis model, why not use silk-ligature for induction of experimental periodontitis? Also, why not include the application of a broad-spectrum antimicrobial agent before OMT?

3. “The use of hydrogel for OMT delivery provides a unique opportunity to seed the micro-organisms onto the surface of teeth for a prolonged period” Any data to support these claims?

4. Safety concerns in the application of OMT are similar to those for oral probiotics, as transplanted biofilms should possess a high degree of genetic stability. It is critical to determine whether oral biofilms should be transplanted directly from donors to patients, or pre-treated with methods that eliminate pathogenic organisms prior to transplantations. How can these problems be solved according to the current protocol?

5. In OMT, transplanted biofilms must exhibit the capacity to effectively endure the selective pressure of the environment. Why do authors choose hydrogels to deliver microbiota for OMT? Please specify in the text.

6. The discussion part should focus more on the advantages of this experimental method

Reviewer #2: The protocol seems to make sense, however the concept of the super donor seems a bit naive at this point. None knows what is the "healthiest" microbiota is and how is it supposed to be characterized. The idea of an in vitro "factory" for the beneficial microbiota is very interesting. The weakness of the protocol is that it artificially induces caries and periodontitis in rats and then attempts "fixing" it. It would be much more interesting to test the OMT with naturally occurring periodontitis in dogs or humans directly. Of note, before COVID, we were able to obtain an IRB approval for a direct human-to-human OMT for volunteer couples. (Our study did not move forward due to COVID) Perhaps the authors of the paper may consider something along these lines as an addition to their protocol. This is because if OMT could be shown to work in principle in a simple setting, there will be a strong motivation to continue with the "super donor".

7. PLOS authors have the option to publish the peer review history of their article (what does this mean?). If published, this will include your full peer review and any attached files.

Reviewer #1: No

Reviewer #2: **Yes: **ALEXANDER POZHIKOV

---

## [Author Response · Author response to Decision Letter 0]

16 Sep 2021

We thank the reviewers for their comments. We have addressed each comment below and in the main text, and we believe that the manuscript now more clearly explains the strategies behind our approach. 

The comments from the reviewer:

This manuscript aims to develop and characterize an Oral Microbiome Transplant (OMT) for the treatment of caries and periodontal disease. Supragingival plaque samples would be collected from healthy participants to find donors suitable for microbial transplantation. Based on the current protocol, a novel in vitro flow cell model combined with microscopic examination and DNA sequencing were be used to describe the composition and functions of the donors. Hydrogels would be used to deliver microbiota for OMT. The effectiveness of OMT would be assessed using caries and periodontitis animal models. Although the current research would provide preliminary results of the feasibility of oral microbiome transplant, I still see some major areas of concern that I think need to be addressed before this manuscript is ready for primetime.

1. What is the specific definition of “super donor”? Authors mentioned that it is prudent to understand the variation and functions of a healthy oral microbiome and hence define a “super donor”, however, authors never presented data in support of any of these claims.

 Author’s response: This concept is borrowed from fecal microbiota transplant literature (reviewed in Wilson, et al. Front. Cell. Infect. Microbiol., 2019). An oral microbiota ‘super donor’ is now defined in the text in the context of this approach and discussed in great detail in the introduction (page 8, Line 169-190) and discussion (page 22, line 486-488, 496-508). 

2. In Rodent periodontitis model, why not use silk-ligature for induction of experimental periodontitis? Also, why not include the application of a broad-spectrum antimicrobial agent before OMT?

 Author’s response: This is a wonderful suggestion. We now provide a rational as to why this approach was not selected, but it would be useful in the exploration of OMT efficacy for a multitude of reasons. We have now included it in the discussion (page 24, line 550-562).

3. “The use of hydrogel for OMT delivery provides a unique opportunity to seed the micro-organisms onto the surface of teeth for a prolonged period” Any data to support these claims?

 Author’s response: As OMT therapy has not yet been delivered yet in humans (which we now make clear in the introduction of the paper), there are not any references to support this claim. Therefore, we have clarified our statement and provided further rational to support the selection of hydrogel developed in OMTs (page 23-24, line 529-546). 

4. Safety concerns in the application of OMT are similar to those for oral probiotics, as transplanted biofilms should possess a high degree of genetic stability. It is critical to determine whether oral biofilms should be transplanted directly from donors to patients, or pre-treated with methods that eliminate pathogenic organisms prior to transplantations. How can these problems be solved according to the current protocol?

 Author’s response: We agree that this is a critical issue. We have added a brief discussion of this into manuscript in the context of our in vivo model (Page 19, line 432-434, and page 20, line 458-462), although directly testing this will likely require early clinical trials in humans. In murine models for caries and periodontitis, prior to application of OMT, full mouth debridement and disinfection would be carried out. For safety assessments in murine models, discomfort would be measured by vital parameters and any adverse reactions (page 20-21, line 477-479). 

5. In OMT, transplanted biofilms must exhibit the capacity to effectively endure the selective pressure of the environment. Why do authors choose hydrogels to deliver microbiota for OMT? Please specify in the text.

 Author’s response: We provide an improved rational for why hydrogels were selected in the discussion ( Page 23-24, line 523-546) and also include a brief discussion of other methods that could be employed, if hydrogels fail (Page 16). We will test the efficacy of delivering OMT in hydrogels in murine models. In addition, other modes of delivery would also be tested such as buffers, varnishes and mouthwash (page 16-17, line 383-386).

6. The discussion part should focus more on the advantages of this experimental method.

Author’s response: We have completely revamped the discussion to include significant discussion of the advantages of this approach (Page 212, line 496-508, page 23 line 520-522, line 523-527, line 540-546, 563-564).

---

## [Decision Letter · Decision Letter 1]

10 Nov 2021

Development and Characterization of an Oral Microbiome Transplant Among Australians for the Treatment of Dental Caries and Periodontal Disease: A Study Protocol.

PONE-D-21-16611R1

Dear Dr. Nath,

We’re pleased to inform you that your manuscript has been judged scientifically suitable for publication and will be formally accepted for publication once it meets all outstanding technical requirements.

Kind regards,

Christopher Staley, Ph.D.

Academic Editor

PLOS ONE

Additional Editor Comments (optional):

Reviewers' comments:

Reviewer's Responses to Questions

**Comments to the Author**

1. Does the manuscript provide a valid rationale for the proposed study, with clearly identified and justified research questions?

Reviewer #2: Yes

Reviewer #3: No

2. Is the protocol technically sound and planned in a manner that will lead to a meaningful outcome and allow testing the stated hypotheses?

Reviewer #2: Yes

Reviewer #3: No

3. Is the methodology feasible and described in sufficient detail to allow the work to be replicable?

Reviewer #2: Yes

Reviewer #3: No

4. Have the authors described where all data underlying the findings will be made available when the study is complete?

Reviewer #2: Yes

Reviewer #3: No

5. Is the manuscript presented in an intelligible fashion and written in standard English?

Reviewer #2: Yes

Reviewer #3: Yes

6. Review Comments to the Author

You may also provide optional suggestions and comments to authors that they might find helpful in planning their study.

Reviewer #2: My major concern was the use of the hydrogel and the definition of the "super donor". Although one may argue if the hydrogel use is warranted, the authors gave a satisfactory rationale. Still, I would leave a room for other potential modes for delivering the OMT other than a hydrogel.

Reviewer #3: The authors are strongly encouraged to perform the proposed study!!!

7. PLOS authors have the option to publish the peer review history of their article (what does this mean?). If published, this will include your full peer review and any attached files.

Reviewer #2: **Yes: **Alexander Pozhitkov

Reviewer #3: No

---

## [Editor Report · Acceptance letter]

15 Nov 2021

PONE-D-21-16611R1 

Development and characterization of an oral microbiome transplant among Australians for the treatment of dental caries and periodontal disease: A study protocol. 

Dear Dr. Nath:

I'm pleased to inform you that your manuscript has been deemed suitable for publication in PLOS ONE. Congratulations! Your manuscript is now with our production department. 

Kind regards, 

on behalf of

Dr. Christopher Staley 

Academic Editor

PLOS ONE